# LAMP: AN LLM-BASED MESSAGE PASSING ARCHITECTURE FOR TEXT-RICH GRAPHS

## ABSTRACT

Text-rich graphs, which integrate complex structural dependencies with abundant textual information, are ubiquitous yet remain challenging for existing learning paradigms. An ideal model must simultaneously satisfy **semantic fidelity** (reasoning over full raw text), **structural integrity** (faithful multi-hop propagation), and **computational scalability** (efficient handling of large neighborhoods). Current approaches inevitably compromise one of these aspects: GNN-based methods compress text into fixed embeddings, losing semantic detail; LLM-based methods serialize graphs into sequences, weakening structural reasoning; and recent "LLM-as-GNN" hybrids improve structural integrity but still bypass explicit reasoning on raw content. We introduce **LAMP**, an **L**LM-based **A**rchitecture for **M**essage **P**assing that overcomes this trade-off. LAMP reinterprets the stacking of decoders as message passing steps and adopts a dual-representation scheme: it anchors inference on each node's raw text during each iteration while propagating compact summaries across neighbors. Furthermore, LAMP unifies discriminative (e.g., node classification) and generative (e.g., GraphQA) tasks under a single generative formulation, allowing end-to-end training without task-specific heads. Extensive experiments show that LAMP effectively unifies graph propagation and text reasoning, achieving competitive performance while offering new insights into the role of LLMs as general-purpose graph learners. *Code will be available upon publication.*

## 1 INTRODUCTION

The pursuit of a model that can master text-rich graphs (Jin et al., 2024; Wang et al., 2025; Zhang et al., 2024a) has led to a frustrating trilemma, forcing a compromise between three essential properties: **Semantic Fidelity** (reasoning over full raw text), **Structural Integrity** (performing faithful multi-hop message passing), and **Computational Scalability**. The failure to resolve this trilemma stems from two foundational architectural bottlenecks. GNN-centric methods (Jin et al., 2024; Wang et al., 2024) create a severe *Semantic Bottleneck*, sacrificing text fidelity by pre-compressing rich text into static embeddings. Conversely, LLM-centric methods (Ye et al., 2023; Chen et al., 2024) create a *Structural Bottleneck*, sacrificing structural integrity by serializing graph topologies into flattened linear sequences. Even recent hybrid models, such as GOFA (Kong et al., 2024) and GL-Fusion (Yang et al., 2024), resort to what we term *invasive integration*. They mechanically inject auxiliary or distinct structure-learning modules into the LLM backbone, creating a disjointed system that decouples graph operations from the LLM's native reasoning.

In contrast to *invasive integration*, the "LLM-as-GNN" paradigm pivots to *behavioral simulation*, where the LLM concurrently executes message passing and semantic reasoning within a unified textual space. As a representative of this approach, PromptGFM (Zhu et al., 2025) coerces an LLM to emulate GNN aggregation using discrete natural language instructions—a technique formalized as hard prompt compression (Li et al., 2025). However, this reliance on discrete text **reintroduces** the *Semantic Bottleneck* in the form of a *Message Bottleneck* during message passing among nodes. By forcing rich neighborhood context through coarse-grained natural language summarization, these models inevitably discard crucial semantic details. Furthermore, this process is inherently unreliable, hinging on the LLM's unpredictable adherence to procedural prompts, and importantly, it lacks an end-to-end mechanism to directly optimize the message representations.

Figure 1: Illustration of message passing paradigms on text-rich graphs. Given a text-rich graph (left) where each node is associated with long textual content, (i) traditional GNNs compress node texts into a compact representation during aggregation, whereas (ii) explicit aggregation paradigm retains original text for LLM reasoning.

However, the mere adoption of continuous representations (soft prompts (Li et al., 2025)), as seen in HiCom (Zhang et al., 2024c), offers only a superficial remedy. While these methods correctly abandon the brittleness of hard prompts, they fail to transcend shallow aggregation because they fall prey to a more fundamental architectural flaw: the *Anchor Node Bottleneck*. This flaw extends to all prior hybrids, including invasive models like GL-Fusion (Yang et al., 2024), as their core aggregation during message passing remains bound to compressed vectors of the target (anchor) node itself. In all such cases, this means the model's most powerful reasoning asset—the anchor node's full, uncompressed raw text—is locked away and inaccessible at the most crucial moment of contextual reasoning. Consequently, the propagation layers operate blindly on condensed states, eroding the semantic fidelity required for deep reasoning. As illustrated in Figure 1 (i), this bottleneck traps existing paradigms in "Compressed Aggregation." To achieve true **Semantic Fidelity**, the LLM must not merely *read* summaries; it must act as the **graph kernel**, with direct, unmediated access to raw content during aggregation.

To overcome these constraints, we propose to **architecturally recast** the LLM to function as such a graph kernel, rather than simply emulating a GNN. We introduce **LAMP** (an LLM-based Architecture for Message Passing), a new paradigm built on the more principled foundation of soft prompt compression. LAMP's core contribution is the architectural reinterpretation of a standard LLM decoder to natively execute a *raw-text-guided* message-passing algorithm. The iterative application of a decoder block becomes a graph propagation hop, and the messages themselves are not discrete text but soft prompts—compact, continuous, and directly optimizable vectors that represent neighbor information.

This paradigm is operationalized through LAMP's dual-representation aggregation scheme, a mechanism that fundamentally distinguishes it from all prior hybrids. At each propagation layer, the update for a target node is uniquely conditioned on two distinct inputs: (1) its own full, uncompressed raw text—thereby bypassing the *Anchor Node Bottleneck*—and (2) the compact soft prompts from its neighbors—mitigating the *Message Bottleneck*. The LLM decoder's attention mechanism becomes the aggregator, grounding its update in rich semantic fidelity while efficiently incorporating structural signals from a large neighborhood. This asymmetric process ensures that, unlike other models that operate on compressed-only representations during propagation, LAMP maintains a direct, persistent link to the target node's raw text at every step. By instantiating GNN message passing as an LLM-native process, LAMP resolves the fidelity-structure-scalability trilemma and provides a more principled blueprint for a true, general-purpose graph learner.

We summarize our contributions as follows:

- We define three essential properties for modeling text-rich graphs—**Semantic Fidelity**, **Structural Integrity**, and **Scalability**—and demonstrate how existing paradigms fail to satisfy them all, creating a fundamental trade-off.

- We propose **LAMP**, a new paradigm for LLM-based message passing that resolves this trade-off through a novel **dual-representation scheme**, enabling both explicit reasoning on a target's raw text and efficient propagation of neighbors' compact summaries.

- We design an adaptive, ratio-based compression mechanism to handle variable text lengths, ensuring scalability while preserving semantics.

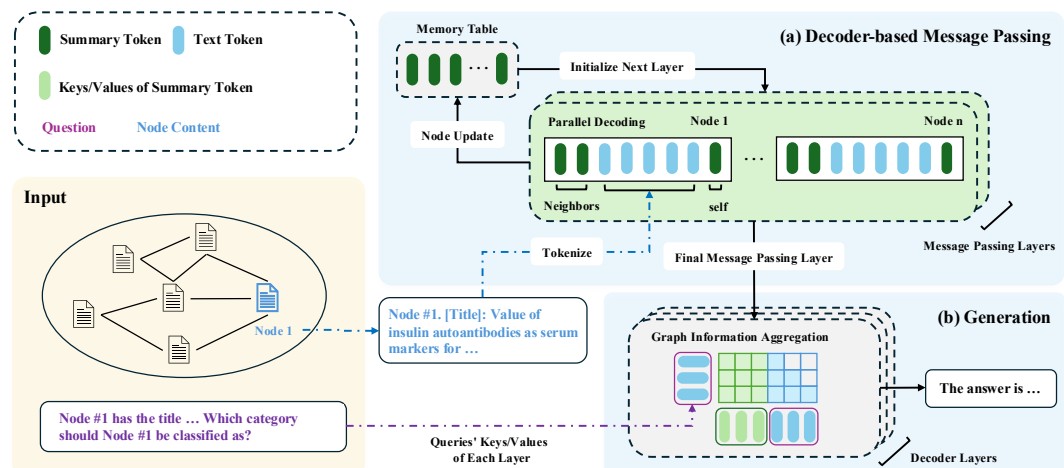

Figure 2: Architecture of LAMP. Given a text-rich graph and a query, we wrap node contents into token sequence and perform parallel decoding on all nodes to obtain their summaries. Hidden states of summary tokens are stored in a memory table to initialize the next layer, realizing (a) layer-wise message-passing; the final decoder aggregates graph information for (b) the answer generation.

- We provide a generative task format for LAMP on both discriminative and generative tasks. Datasets, code, and models will be released upon acceptance.

A more comprehensive discussion of related work is provided in App. D.

## 2 METHOD

In this section, we present the LAMP framework. We begin by revisiting the classical GNN paradigm to identify the primary challenge in applying it to text-rich graphs. We then argue that LLMs, particularly their inherent capacity for context compression, offer a natural and powerful solution to this challenge. Finally, we introduce LAMP as an *LLM-native* GNN counterpart, detailing its architecture and training recipe.

### 2.1 REVISITING GNNS AND THE TEXT-RICH BOTTLENECK

The core of modern GNNs is the **message passing** paradigm, where each node updates its representation by aggregating information from its local neighborhood. Following the general formulation that unifies models like MPNN (Cai et al., 2023) and GAT (Veličković et al., 2017), a GNN layer is defined as:

$$\mathbf{h}_i^{(l+1)} = \mathbf{Update}\Big(\mathbf{h}_i^{(l)}, \mathbf{Aggregate}(\{\mathbf{Msg}(\mathbf{h}_j^{(l)}) : j \in \mathcal{N}(i)\})\Big), \tag{1}$$

where $\mathbf{h}_i^{(l)}$ is the vector representation (or state) of node $i$ at layer $l$, and $\mathcal{N}(i)$ is its neighbor set. This formulation reveals a fundamental asymmetry in how GNNs treat the target node versus its neighbors. This process involves three key functions: 1. 'Msg': Transforms a neighbor's state $\mathbf{h}_j^{(l)}$ into a message. 2. 'Aggregate': A permutation-invariant function (e.g., sum, mean, or attention) that combines messages from all neighbors into a single vector. 3. 'Update': Fuses the aggregated message with the target node's own state $\mathbf{h}_i^{(l)}$ to produce the new state $\mathbf{h}_i^{(l+1)}$. Often, this step includes a residual or skip connection to $\mathbf{h}_i^{(l)}$ to preserve the node's identity and stabilize training in deep models. A crucial insight from this formulation is the **asymmetric treatment** of the target node versus its neighbors. The target node's state, $\mathbf{h}_i^{(l)}$, is preserved as a distinct **self-representation** that guides the aggregation. Taking GAT (Veličković et al., 2017) as an example, $\mathbf{h}_i^{(l)}$ acts as a **query** to compute attention weights over its neighbors' states, which serve as **keys** and **values**. This allows the model to learn which neighbors are most relevant, a design that significantly boosts performance by enabling focused, context-aware updates. While powerful, this paradigm faces a fundamental **infor-**

**mation bottleneck** when applied to text-rich graphs. GNNs operate on fixed-size vectors ($\mathbf{h}_i \in \mathbb{R}^d$), forcing any textual attributes to be pre-compressed into a single embedding before message passing begins. This premature compression of long, nuanced documents into a single vector inevitably discards critical semantic details, a problem well-documented in the literature (Zhang et al., 2024c). Consequently, the GNN's message-passing layers never reason over the full raw text, fundamentally limiting the model's semantic fidelity. This begs the question: how can we build a GNN counterpart that performs message passing directly on raw text without sacrificing scalability?

## 2.2 LLMs as Natural Graph Propagators

We find a compelling answer in recent advancements in long-context LLMs. These models have faced a parallel challenge: the quadratic complexity of self-attention limits their context window. A highly successful solution has been *context compression*, where the LLM itself is adapted to distill long sequences of text into a few compact, special-purpose tokens. These tokens, referred to as "summary vectors" (Chevalier et al., 2023), "memory slot" (Ge et al., 2023), or "activation beacons" (Zhang et al., 2024b), function as learned, compact representations—or embeddings—of the original text. Crucially, models like ICAE (Ge et al., 2023) and AutoCompressor (Chevalier et al., 2023) have shown that an LLM can be trained via a simple auto-encoding objective to generate these summary tokens, from which the original text can be faithfully reconstructed. This demonstrates that an LLM is a powerful and natural tool for information-preserving compression. This principle of LLM-driven context compression perfectly fills the gap in text-rich GNNs. We propose to re-imagine the GNN message-passing functions ('Msg', 'Aggregate', 'Update') through the lens of an LLM decoder. The LLM itself becomes the propagation engine, performing all GNN operations at the token level. To this end, we introduce LAMP, which establishes a direct correspondence between classical GNN components and LLM-native operations. Specifically, we make the following critical mapping: (1) The target node's high-fidelity state $\mathbf{h}_i^{(l)}$ is instantiated as its **raw text,** $\mathbf{X}_i$. (2) The messages from neighbors, $\mathbf{Msg}(\mathbf{h}_j^{(l)})$, are instantiated as compact **summary tokens,** $\mathbf{S}_j^{(l)}$. By leveraging an LLM decoder as the core computational unit, LAMP creates an elegant and powerful framework for scalable, structurally faithful learning on text-rich graphs.

## 2.3 LAMP: LLM-based Architecture for Message Passing

The LAMP framework shown in Fig. 2 consists of three main stages, mirroring the structure of GNN-based learning: (1) initializing node representations, (2) performing multi-hop message passing, and (3) making a final prediction. For clarity, we use $\mathbf{h}_i^{(l)}$ to denote the state of node $i$ at layer $l$ in a classical GNN and $\mathbf{S}_i^{(l)}$ for its counterpart in LAMP (i.e., summary tokens).

**Step 1: Message Initialization via Text Compression.** In classical GNNs, the process begins by creating an initial state $\mathbf{h}_i^{(0)}$ for each node. In LAMP, we achieve the equivalent by generating an initial sequence of summary tokens $\mathbf{S}_i^{(0)}$ for every node $i$ by compressing its raw text $\mathbf{X}_i$. This step can be viewed as creating the initial "messages" that each node is capable of sending. Inspired by parallel-decoding techniques (Zhang et al., 2024b; Li et al., 2024b), we prompt an LLM decoder to perform this compression. Concretely, for each node $i$, we insert corresponding summary-token placeholders into the input text $\mathbf{X}_i$, and let the decoder generate hidden states for these special tokens: $\mathbf{S}_i^{(0)} = [s_{i,1}^{(0)}, \ldots, s_{i,n}^{(0)}]$, $n = \lceil \rho \cdot L_i \rceil$, where $\rho \leq 1$ is a predefined compression ratio. The number of summary tokens, $n$, is proportional to the original text length $L_i$ of $\mathbf{X}_i$, allowing the model to allocate more capacity to richer content. These summary tokens $\mathbf{S}_i^{(0)}$ are the *LLM-native* equivalent of initial node embeddings, but are variable-length and information-rich.

**Step 2: Message Passing via Parallel Decoding.** This step unifies the GNN's 'Aggregate' and 'Update' functions into a single, parallelizable decoder inference step (as shown in Fig. 2(a)). At each layer $l$, LAMP updates the summary tokens for all nodes. For a target node $i$, the decoder takes a single input sequence constructed by concatenating three distinct sources of information:

$$\mathbf{S}_i^{(l+1)} = \mathbf{Decoder}\Big( \big[ \underbrace{\mathbf{S}_{j_1}^{(l)} \| \cdots \| \mathbf{S}_{j_m}^{(l)}}_{\text{neighbors}} \| \underbrace{\mathbf{X}_i}_{\text{raw text}} \| \underbrace{\mathbf{S}_i^{(l)}}_{\text{self}} \big] \Big), j_1, \ldots, j_m \in \mathcal{N}(i). \tag{2}$$

This design directly operationalizes the asymmetric GNN principle within an LLM: **(a) Neighbor Messages:** The summary tokens of neighbors, $\{\mathbf{S}_j^{(l)}\}$, are concatenated to form the aggregated neighborhood information, corresponding to $\mathbf{Aggregate}(\{\mathbf{Msg}(\mathbf{h}_j^{(l)})\})$. **(b) Self-State (Query):** The target node's **raw text**, $\mathbf{X}_i$, represents its full, high-fidelity state, $\mathbf{h}_i^{(l)}$. It acts as a powerful, dynamic query that allows the decoder's self-attention to determine which parts of the neighbor messages are most relevant to the node's complete semantic content. **(c) Self-State (Prompt/Skip):** The target node's own summary tokens from the previous layer, $\mathbf{S}_i^{(l)}$, are also included. This serves a dual role. Functionally, it acts as a **skip connection**, ensuring that the model can carry forward its previously learned representation. Mechanistically, by placing it at the end of the input sequence—after the neighbor summaries and the node's own raw text—it serves as the direct prompt for the autoregressive decoder. A decoder generates an output sequence by predicting one token at a time, conditioned on a preceding context or prompt. To generate the *new* summary sequence $\mathbf{S}_i^{(l+1)}$, we use the *current* summary tokens $\mathbf{S}_i^{(l)}$ as the generative prompt. The decoder is thus tasked with "editing" or "rewriting" its input summary based on the provided context. This turns message passing into a sophisticated in-context revision task, where the LLM refines a node's summary by integrating neighborhood information under the guidance of its own full-text semantics. The decoder processes this structured input and outputs a new sequence of hidden states, which constitute the updated summary tokens $\mathbf{S}_i^{(l+1)}$. These states are then stored in a memory table $\mathcal{M}$ to serve as messages for the next layer of propagation.

**Step 3: Generative Task Formulation.** After $L$ rounds of message passing, LAMP uses the final summary token representations to perform downstream tasks. Unlike classical GNNs that require task-specific heads, LAMP unifies all tasks under a single **generative formulation** (see Fig. 2(b)). Here, the role of the summary tokens shifts from being *states-to-be-updated* to being *context-to-be-reasoned-with*. This is where the Key-Value (KV) cache becomes essential. The final layer summary tokens, $\mathbf{S}_i^{(L)}$, provide two complementary forms of information: **1. Hidden States:** The final representations stored in the memory table, which embody the propagated node state. **2. Key-Value (KV) Cache:** Derived from these hidden states, the KV pairs are the mechanism by which the decoder can efficiently attend to the full graph context during auto-regressive generation. **This deliberate separation of state propagation and final generation is a key design choice.** During the iterative message passing in Step 2, we only propagate the compact hidden states. This is for two reasons. First, it mirrors how classical GNNs propagate state vectors layer by layer, where the goal is to compute the next layer's representation, not generate a final, human-readable answer. Second, it is significantly more memory-efficient. Storing the full KV cache for every node at every intermediate GNN layer would be prohibitively expensive, especially for deep LLMs and large graphs. The KV cache is only materialized at the very end, providing the final decoder with the necessary context for rich, generative reasoning without the massive overhead during propagation. This approach is inspired by modern long-context models like FocusLLM (Li et al., 2024b), which separate the creation of compressed context from its utilization in a final decoding pass. For a given task query $Q$ and a target answer $Y = (y_1, \ldots, y_T)$, we feed the aggregated KV caches from the relevant nodes into the final decoder. Here the same decoder is used for both message passing and generation, with shared parameters. The model is then trained end-to-end to maximize the conditional probability:

$$\mathcal{L}_{\text{gen}} = -\sum_{t=1}^{T} \log P(y_t \mid y_{<t}, Q, \mathbf{K}, \mathbf{V}), \tag{3}$$

where $(\mathbf{K}, \mathbf{V})$ are the final key-value caches. This approach is highly flexible: for node-level tasks, only the target node's KV cache is used; for graph-level tasks, the KV caches of all relevant nodes are provided, analogous to graph pooling in classical GNNs.

**Discussion.** Our proposed LAMP architecture can be viewed as an *LLM-style* generalization of the classical GNN message-passing paradigm for text-rich graphs, as summarized in Tab. 1. A key advantage of LAMP is its **fidelity-preserving message passing**. It achieves this by using a node's full, raw text ($\mathbf{X}_i$) to guide the attention over its neighbors' summary tokens $\mathbf{S}_j$, rather than operating solely on compressed embeddings $\mathbf{h}_j$ as in GNNs. This avoids the irreversible semantic distortion inherent in traditional pipelines. Consequently, instead of a simple aggregation function (e.g., mean or sum), LAMP employs an attention mechanism over the concatenated sequence through an LLM decoder, turning aggregation into an inference task.

Table 1: Connection between classical GNN message passing and the proposed LAMP.

| GNN operation | LAMP counterpart |
|---|---|
| Node representation $\mathbf{h}_i^{(l)}$ | Summary token sequence $\mathbf{S}_i^{(l)} = [s_{i,1}^{(l)}, \ldots, s_{i,n}^{(l)}]$ |
| Neighbor aggregation $\{\mathbf{h}_j^{(l)} : j \in \mathcal{N}(i)\}$ | Concatenation of neighbor tokens $\mathbf{S}_j^{(l)}$ for $j \in \mathcal{N}(i)$ |
| Update function $\text{GNN}(\mathbf{h}_i^{(l)}, \{\mathbf{h}_j^{(l)}\})$ | LLM reasons over $[\mathbf{S}_{j_1}^{(l)}\|\cdots\|\mathbf{S}_{j_m}^{(l)}\|\mathbf{X}_i\|\mathbf{S}_i^{(l)}]$ |
| Layer-wise propagation | Updating the memory table with summary tokens $\mathbf{S}_i^*$ |

## 2.4 TRAINING RECIPE

LAMP is trained in two consecutive stages: large-scale pre-training and task-specific fine-tuning.

**Pre-training.** Pre-training aims to teach the model two fundamental skills: text compression and message passing. We use two reconstruction-based tasks: **1. Self-reconstruction:** The model must reconstruct a node's raw text $\mathbf{X}_i$ given only its own summary tokens $\mathbf{S}_i$. This trains the compression mechanism to be information-preserving, akin to the objective in FocusLLM (Li et al., 2024b). **2. Neighbor-reconstruction:** The model must reconstruct $\mathbf{X}_i$ given the concatenated summary tokens from its one-hop neighborhood, $\{\mathbf{S}_j : j \in \mathcal{N}(i)\}$. This supervises the message-passing ability, encouraging that summaries contain sufficient information for a neighbor to infer content. At each training step, one task is uniformly selected to reconstruct a randomly chosen target node $i$ 's raw text $\mathbf{X}_i$. During pre-training, we only update the lightweight attention layers related to the summary tokens, following efficient long-context fine-tuning practices (Zhang et al., 2024b; Li et al., 2024b). Further details on the pre-training setup are provided in App. A.1.

**Fine-tuning.** After pre-training, LAMP is adapted to downstream tasks using the same generative formulation in equation 3 and fine-tuned efficiently with LoRA (Hu et al., 2022). A task-specific query (e.g., "Classify this node") is provided, and the model is trained to generate the desired output (e.g., "The category is Machine Learning"). This unified framework allows LAMP to seamlessly handle both discriminative and generative graph learning tasks.

## 3 EXPERIMENT

This section evaluates LAMP by systematically investigating its performance against the three desiderata established in the introduction: **Q1 (Semantic Fidelity):** Does LAMP preserve semantic fidelity and enable effective message passing during pre-training? **Q2 (Structural Integrity):** Does LAMP maintain structural integrity through multi-hop message passing on downstream tasks? **Q3 (Computational Scalability):** Can LAMP scale with increasing graph size without collapsing performance?

## 3.1 LAMP PRE-TRAINING: VALIDATING SEMANTIC FIDELITY

To answer **Q1**, we evaluate the quality of the summary tokens generated during pre-training. We pre-train LAMP on Qwen-2.5-7B-Instruct (Team, 2024), following the training recipe in Sec. 2.4, and then measure its reconstruction perplexity on the unseen Cora dataset. As shown in Tab. 2, we report results under two reconstruction modes: self-reconstruction (*self*), where a node's text is recovered from the key–values of its own summary tokens, and neighbor-reconstruction (*nbr*), where the text is

Table 2: Evaluation for pre-trained LAMP. Qwen-2.5-7B-Instruct (21.72*) is the backbone reference; its value is the raw language modeling perplexity obtained by directly continuing node text.

| | Perplexity$_{self}$ $\downarrow$ | Perplexity$_{nbr}$ $\downarrow$ |
|---|---|---|
| Qwen-2.5-7B-Instruct | 21.72* | - |
| LAMP$_{mp=1,\rho=0.05}$ | 11.36 | 20.68 |
| LAMP$_{mp=1,\rho=0.1}$ | 9.17 | 20.87 |
| LAMP$_{mp=2,\rho=0.1}$ | 8.93 | 22.97 |

generated based on the summary tokens of its neighbors. We further compare different strategies of message-passing rounds ($mp$) and compression ratios ($\rho$) to examine how deeper propagation and richer summaries affect reconstruction quality.

The results in Tab. 2 validate the core mechanics of LAMP. First, the low perplexity in self-reconstruction confirms that LAMP's summary tokens achieve high semantic fidelity, effectively

compressing raw text into a rich representation with minimal information loss (e.g., 9.17 vs. 21.72 for the backbone). Second, the strong performance on neighbor-reconstruction demonstrates that these compact summaries serve as effective messages, as a node can reconstruct its neighbor's content from the information propagated across the graph. This confirms that message passing is successful.

Moreover, we observe a fascinating trade-off with deeper propagation. While an additional message-passing round ($mp = 2$) further improves self-reconstruction, it slightly increases perplexity in the neighbor-reconstruction task. This suggests that as messages propagate, the summary tokens evolve from pure compression to more abstract, contextually-integrated representations. These 2-hop summaries are less suited for verbatim reconstruction of a single neighbor but, as we will see in Sec. 3.2, are more powerful for high-level downstream tasks like classification.

## 3.2 STRUCTURAL AWARENESS: FROM NODE CLASSIFICATION TO STRUCTURE SENSITIVITY

To answer **Q2**, we assess whether LAMP exploits graph structure from the following three aspects: (i) performance on foundational text-rich graph tasks, (ii) performance under different message-passing rounds, and (iii) response to structural perturbations (neighbor-order permutation and connectivity perturbation).

### 3.2.1 FOUNDATIONAL GRAPH TASKS

**Node Classification.** We evaluate LAMP on standard node classification benchmarks, comparing against GNNs (GCN, GAT, GraphSAGE), PLMs (BERT, RoBERTa, Qwen-2.5-7B-Instruct), GNN-LLM Integration methods (OFA and LLaGA), and a recent attempt to make LLMs function as GNNs (PromptGFM).

As shown in Tab. 3, LAMP-7B consistently outperforms both traditional GNNs and text-only PLM baselines across all datasets. This dual advantage stems directly from its unique architecture, which successfully unifies the strengths of both paradigms. Against GNNs like GraphSAGE, LAMP's superiority comes from its preservation of **Semantic Fidelity**; traditional GNNs suffer an information bottleneck by pre-compressing rich text into fixed embeddings, whereas LAMP's message passing layers reason over the target node's full raw text to capture nuanced semantics. Simultaneously, against PLMs like its own backbone (Qwen-2.5-7B-Instruct), LAMP's gains demonstrate the value of **Structural Integrity**. While the backbone LLM only processes isolated node texts, LAMP's message passing allows it to integrate rich contextual information from the multi-hop neighborhood for more informed predictions. By unifying deep text reasoning with graph propagation, LAMP thus overcomes the limitations of both prior approaches. Crucially, the architectural unification also allows LAMP to consistently surpass OFA and LLaGA across most benchmarks, achieving leads up to 6.77% (on PubMed) and 4.83% (on Arxiv), validating the proposed message-passing scheme.

Table 3: Performance comparison of different methods on node classification tasks (Accuracy % ↑). *Results of PromptGFM are quoted from its original paper. Since PromptGFM relies on GPT-4o (closed-source model) and may use different data splits, these numbers are not strictly comparable and are provided for reference only. We highlight in bold the best results among baselines that we directly reproduce under our setting.

| Method | Cora | Citeseer | PubMed | History | Photo | Arxiv |
|---|---|---|---|---|---|---|
| MLP | 69.00 | 59.35 | 75.55 | 77.71 | 47.56 | 57.23 |
| GCN | 82.29 | 70.16 | 81.92 | 80.55 | 70.14 | 66.52 |
| GAT | 83.03 | 71.29 | 80.78 | 78.93 | 66.18 | 67.76 |
| GraphSAGE | 83.21 | 70.81 | 82.94 | 80.72 | 73.82 | 67.97 |
| BERT | 80.99 | 71.93 | 91.75 | 80.94 | 58.34 | 65.64 |
| RoBERTa | 76.93 | 70.48 | 91.37 | 79.42 | 57.27 | 66.72 |
| Qwen-2.5-7B-Instruct | 82.65 | 74.35 | 90.79 | 84.17 | 75.12 | 72.99 |
| OFA | 81.73 | 74.19 | 86.91 | 81.75 | **77.23** | 64.93 |
| LLaGA | 82.28 | 73.54 | 83.89 | 82.54 | 75.15 | 70.55 |
| PromptGFM* | 92.42 | 85.32 | 94.65 | 86.72 | 86.61 | 83.78 |
| LAMP-7B | **84.87** | **74.83** | **93.68** | **85.09** | 76.21 | **75.38** |

Finally, we include results of PromptGFM (Zhu et al., 2025), which reports state-of-the-art (SOTA) scores by leveraging GPT-4o as its graph understanding module. While not directly comparable, these numbers serve as a reference upper bound. Importantly, despite being built on a 7B open-source backbone, LAMP approaches the performance of this GPT-4o–based system. The gap is only 0.97% on PubMed and 1.63% on History, while averaging about 8% lower across all six benchmarks, showing the competitiveness of lightweight, reproducible alternatives. Crucially, unlike PromptGFM which operates on pre-summarized text, LAMP's message passing layers reason directly over the target node's raw text at each step, maintaining full semantic fidelity throughout the aggregation process.

**GraphQA.** Beyond node classification, we also evaluate LAMP on the GraphQA task, where the graph offers additional contextual information for answering questions. Tab. 4 compares LAMP with its backbone, Qwen-2.5-7B-Instruct, and a GraphRAG baseline G-Retriever (He et al., 2024).

Table 4: Evaluation of LAMP on GraphQA tasks (Accuracy % ↑).

| Method | ExplaGraphs |
|---|---|
| Qwen-2.5-7B-Instruct | 93.50 |
| G-Retriever | 87.05 |
| LAMP-7B | **93.86** |

Table 5: Impact of message-passing rounds (Accuracy % ↑).

| Method | Cora | PubMed |
|---|---|---|
| Qwen-2.5-7B-Instruct | 82.65 | 90.79 |
| $\text{LAMP}_{mp=1,\rho=0.1}$ | 83.21 | 92.11 |
| $\text{LAMP}_{mp=2,\rho=0.1}$ | **84.87** | **93.68** |

According to the results, LAMP-7B achieves a 0.36% improvement over the backbone, and also surpasses the GraphRAG baseline G-Retriever, indicating that message passing allows the model to better leverage complementary signals from the graph. Although the gain against the backbone is modest, this result validates that LAMP's architecture extends beyond classification and can incorporate structural context in generative reasoning tasks.

### 3.2.2 IMPACT OF MESSAGE-PASSING ROUNDS

To examine the impact of message-passing rounds, we conduct corresponding experiments on datasets from two different domains: **Cora** (Computer Science) and **PubMed** (Biomedical).

As shown in Tab. 5, simply using Qwen-2.5-7B-Instruct backbone without any message passing achieves 82.65% on Cora and 90.79% on PubMed. With LAMP, one round of message passing ($mp = 1$) already brings clear improvements (increasing to 83.21% on Cora and 92.11% on PubMed). When we further extend the message-passing rounds to two rounds ($mp = 2$), the accuracy improves further, by 1.66% on Cora and 1.57% on PubMed. These results suggest that LAMP benefits from deeper propagation, a phenomenon similar to traditional GNNs: multiple rounds of message passing allow nodes to integrate richer contextual signals from their multi-hop neighborhoods, leading to consistent gains across domains.

### 3.2.3 PERMUTATION INVARIANCE AND STRUCTURE SENSITIVITY

In this section, we design two shuffling tests to verify the typical properties expected from a graph-aware model: (i) **insensitivity to neighbor order** (Buterez et al., 2022), where predictions should remain stable under permutations of neighbors, and (ii) **sensitivity to the graph structure** (Guan et al., 2025), where performance should degrade if the original structural consistency is disrupted.

Correspondingly, the experimental settings are: **(1) Neighbor-Order Shuffle (Permutation).** For each node in the graph, we randomly permute the order of its one-hop neighbors while keeping their contents unchanged. This tests whether LAMP is invariant to neighbor ordering. **(2) Cross-Node Shuffle (Break Structure).** For each graph sample, we randomly swap the neighbor sets between different nodes, so that each node still remains on the graph, but its neighborhood no longer corresponds to the original graph structure. This evaluates whether LAMP exploits signals from the graph structure in addition to semantic information.

The results in Tab. 6 provide definitive evidence of LAMP's structural integrity. First, performance remains stable when the order of neighbors is shuffled (Neighbor-Order Shuffle), confirming that LAMP learns a **permutation-invariant** aggregation function, a core property of GNNs. Second, performance degrades noticeably when the graph's topology is broken (Cross-Node Shuffle),

demonstrating that LAMP is not a mere "bag-of-neighbors" model but derives meaningful signals from the graph's specific connectivity.

Table 6: Impact of Shuffling (Accuracy % ↑). Each shuffling experiment is repeated three times and we report mean and standard deviation.

| Setting | Perturbation Type | Cora | Citeseer |
|---|---|---|---|
| LAMP (No Shuffle) | - | 84.87 | 74.83 |
| Neighbor-Order Shuffle | Permutation | $84.81 \pm 0.08$ | $74.73 \pm 0.01$ |
| Cross-Node Shuffle | Break Structure | $84.32 \pm 0.14$ | $74.41 \pm 0.01$ |

Collectively, the strong performance on downstream graph tasks (Sec. 3.2.1), the consistent gains from deeper propagation (Sec. 3.2.2), and the principled behavior in shuffle tests (Sec. 3.2.3) provide comprehensive evidence that LAMP successfully maintains structural integrity, operating as a true graph-aware learning architecture.

### 3.3  SCALABILITY OF LAMP WITH GRAPH SIZE

To answer **Q3**, we examine whether LAMP can scale with increasing graph size. We group subgraph samples (see App. A.2 for subgraph construction) into different size buckets, reporting the corresponding node classification accuracy and inference time across them in Fig. 3. Although the subgraphs are of moderate size, our largest buckets already reach up to 100 nodes. This is made possible by LAMP's design: whereas prior

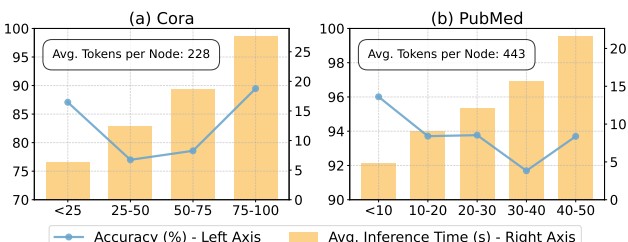

Figure 3: Accuracy and inference time of LAMP across different subgraph sizes on Cora and PubMed; the x-axis denotes the number of nodes in each graph.

"LLM-as-GNN" approaches often face quadratic complexity when concatenating node texts into a single sequence (e.g., PromptGFM), LAMP assigns each node to a single decoder and propagates only compact summaries. This reduces the cost of message passing from quadratic to linear in the number of nodes while retaining semantic fidelity, enabling us to handle substantially larger subgraphs in practice. The results show that LAMP maintains stable accuracy as graph size increases, with no clear degradation in performance.

## 4  CONCLUSION

In this paper, we introduced LAMP, an LLM-based architecture for message passing that unifies text reasoning with graph propagation through a dual-representation scheme. LAMP achieves semantic fidelity, structural integrity, and computational scalability, demonstrating competitive performance on different downstream tasks. Interestingly, we also find that intermediate message-passing rounds can already produce meaningful generations (App. F). This behavior further supports interpreting LAMP as a potential form of multi-agent communication (Zhuge et al., 2024; Qian et al., 2024), where each decoder acts as an LLM agent exchanging information through message passing.

**Limitations.** LAMP's ratio-based compression already adapts to variable text lengths, but remains heuristic and may be suboptimal across tasks. Additionally, we find 2-hop reasoning empirically sufficient for current benchmarks, leaving the exploration of deeper propagation for future work. While our current work establishes a strong baseline on homogeneous graphs, promising future directions include: (1) assessing model robustness in heterogeneous, noisy, and multilingual settings; (2) developing adaptive compression strategies; and (3) expanding to broader applications.

## ETHICS STATEMENT

Our research is based on public benchmark datasets, which involve no human subjects or sensitive data. We foresee no resulting ethical concerns or negative societal impact.

## REPRODUCIBILITY STATEMENT

To ensure reproducibility, we provide comprehensive details in the Appendix, including our experimental setup (App. A), and dataset pre-processing steps (App. B).

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

APPENDIX

## A    IMPLEMENTATION DETAILS

We provide detailed descriptions of our implementation in this section. Specifically, we divide it into three parts: (1) settings for pre-training, where we describe the backbone model, optimization strategy, and training environment; (2) settings for downstream tasks, including fine-tuning configurations and prompt templates; and (3) training data construction, covering how we prepare both pre-training corpora and task-specific benchmarks.

### A.1    SETTINGS FOR PRE-TRAINING

We adopt Qwen-2.5-7B-Instruct (Team, 2024) as the backbone and follow the practice of Focus-LLM (Li et al., 2024b) and Activation Beacon (Zhang et al., 2024b) by freezing all backbone parameters and fully fine-tuning the additional attention layer. Training is conducted on 32 NVIDIA A100 GPUs for one epoch over the entire pre-training corpora (see App. A.3 for data construction). During pre-training, we adopt the optimization hyperparameters recommended by Li et al. (2024b), using BF16 mixed precision for training.

### A.2    SETTINGS FOR DOWNSTREAM TASKS

For fine-tuning on node-level tasks, we adopt the standard practice of constructing subgraph inputs (Kong et al., 2024). Specifically, for a given target node, we expand its neighborhood to form an ego-graph, which is then fed into the model. To control input length, we set a maximum subgraph size and randomly subsample neighbors when the expansion exceeds this threshold. Unless otherwise specified, all downstream experiments are conducted with the LAMP model pre-trained on Qwen-2.5-7B-Instruct, using two message-passing rounds and a compression ratio of $\rho = 0.1$.

During fine-tuning, we apply LoRA with rank $= 64$, $\alpha = 64$, and dropout $= 0.1$. We use early stopping with a patience of 3 epochs, based on validation accuracy. The prompt templates we use for fine-tuning are displayed in Sec. E. All experiments are conducted on 16 NVIDIA A100 GPUs.

### A.3    TRAINING DATA CONSTRUCTION

For pre-training, we construct corpora from the graph format of the MAPLE dataset (Zhang et al., 2023), a large-scale dataset for scientific literature tagging. To ensure domain diversity while avoiding data leakage into downstream evaluations, we sample subgraphs from the following three domains: **Economics**, **Mathematics**, and **Geology**. In total, we generate over **80,000** subgraph samples. Each subgraph is obtained by randomly selecting a central node and then expanding along its edges to form a small ego-graph. The subgraph size is uniformly sampled between 20 and 80 nodes, so that the model is exposed to graphs of varying scales while maintaining training efficiency.

Each node retains its raw textual content (i.e., paper title and abstract) as well as connectivity. As described in Sec. 2.4, two reconstruction tasks are adopted during pre-training: (1) **self-reconstruction**, which performs reconstruction through the node's own summary tokens, and (2) **neighbor-reconstruction**, which conditions on the summary tokens of one's neighbors when available, defaulting to self-reconstruction otherwise.

Fine-tuning is performed on task-specific benchmark datasets (e.g., Cora, Citeseer, PubMed), which are described in App. B.

## B    DATASET

We evaluate LAMP on multiple benchmark datasets spanning different domains and tasks. Specifically, we consider (a) six node classification datasets that vary in domain and graph density, and (b) a GraphQA dataset for generative reasoning with structured evidence.

## B.1 Node Classification

Our experiments are conducted on six benchmark node classification datasets. Tab. 7 presents a statistical summary of these datasets, including their graph properties, domain, and the type of node text. For consistency, all graphs are treated as undirected. We elaborate on each dataset below.

Table 7: Dataset Statistics. "Avg. D" means the average node degree. We additionally provide the domain and the composition of node text.

| Dataset | #Nodes | #Edges | #Classes | Avg. D | Domain | Node Text |
|---|---|---|---|---|---|---|
| Cora | 2,708 | 10,858 | 7 | 4.01 | Computer Science | Paper titles and abstracts |
| Citeseer | 3,327 | 9,464 | 6 | 2.84 | Computer Science | Paper titles and abstracts |
| PubMed | 19,717 | 88,648 | 3 | 13.77 | Biomedical | Paper titles and abstracts |
| History | 41,551 | 503,180 | 12 | 18.07 | E-commerce | Item titles and descriptions |
| Photo | 48,362 | 873,782 | 12 | 12.11 | E-commerce | Item titles and reviews |
| Arxiv | 169,343 | 1,166,243 | 40 | 6.89 | Computer Science | Paper titles and abstracts |

- **Cora (McCallum et al., 2000).** In this dataset, nodes represent scientific publications in the machine learning domain, and edges represent citation links between them. The task is to classify each publication into one of seven predefined subject categories (e.g., Neural Networks, Reinforcement Learning). We further divided these nodes into training, validation, and test subsets, using a 7:1:2 ratio.

- **Citeseer (Giles et al., 1998).** The Citeseer dataset is another widely-used citation network. Similar to Cora, nodes represent scientific publications and edges represent citations. The classification task is to assign each publication to one of six categories (e.g., AI, DB, IR). We further divided these nodes into training, validation, and test subsets, using a 7:1:2 ratio.

- **PubMed (Sen et al., 2008).** The PubMed dataset is a larger citation network sourced from the PubMed medical database. Nodes represent scientific papers, and edges represent citation links. The task is to classify each paper into one of three types. We further divided these nodes into training, validation, and test subsets, using a 7:1:2 ratio. To improve computational efficiency for models, we further subsampled the training and validation sets to 10,000 and 300 nodes, respectively. The smaller validation set was used for early stopping to prevent overfitting during training.

- **History (Yan et al., 2023).** In this graph, nodes represent individual books, and an edge connects two nodes if the corresponding books are frequently co-purchased or co-viewed by customers. Node text is the book's title and description. The node classification task is to predict the category of each book from a set of 12 distinct classes. Following the same protocol as with the PubMed dataset, we utilize a subsampled training/validation set.

- **Photo (Yan et al., 2023).** The Amazon Photo dataset is a co-purchase network from the Amazon-Electronics domain, where nodes represent products and edges indicate frequent co-purchases or co-views. Node text is based on textual reviews, and the task is to classify products into 12 categories. We employ the same data partitioning and training set subsampling strategy for this dataset as described for PubMed and History.

- **Arxiv (Hu et al., 2020).** The OGBN-Arxiv dataset is a computer science citation network collected from the Arxiv platform, where nodes represent scientific papers and edges indicate citation links between them. Node text is typically based on the paper title and abstract. The task is to classify each paper into one of 40 predefined subject categories. Based on the original data split, we randomly sample *10,000*, 2,000, and 10,000 target nodes for training, validation, and testing, respectively. Crucially, while the supervision signals are downsampled, all applicable models retain access to the complete underlying graph topology for neighbor aggregation, ensuring structural integrity is preserved.

## B.2 GraphQA

We also conduct experiments on the following GraphQA dataset.

- **ExplaGraphs (Saha et al., 2021).** This dataset is designed for generative commonsense reasoning. It focuses on creating explanation graphs to facilitate stance prediction in debates. The core task is to predict whether arguments are supportive or contradictory to a given belief, with Accuracy serving as the primary metric. The graphs in this dataset contain rich semantic information on both nodes and edges. To fully leverage this, we transform the original graphs by converting each edge into a node (as depicted in Fig. 4), subsequently creating a bipartite graph structure. We follow the data split setting from G-Retriever (He et al., 2024), partitioning the dataset into training, validation, and test sets with a ratio of 6:2:2.

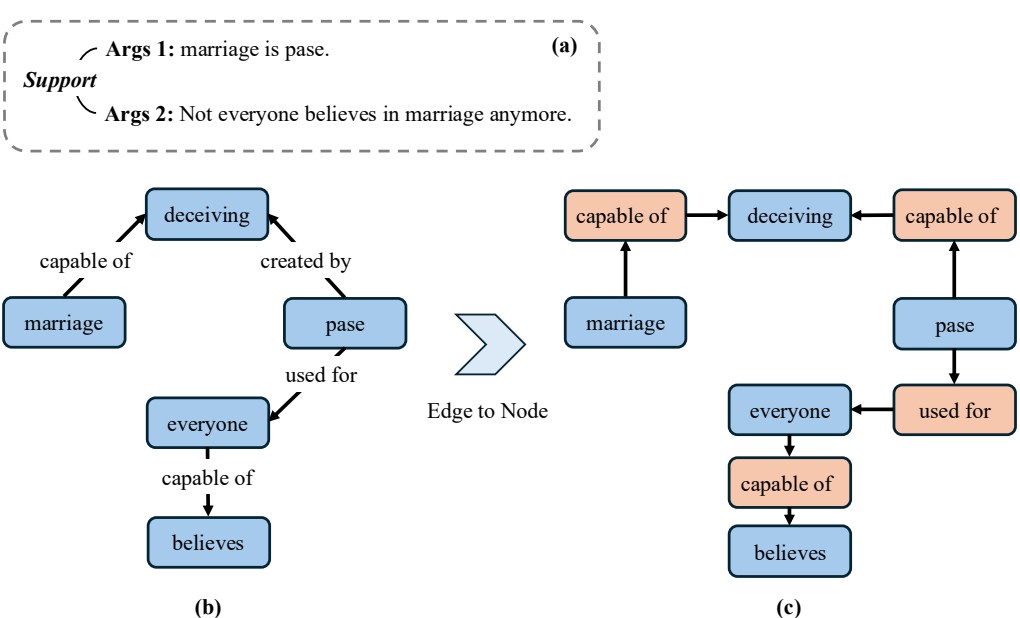

Figure 4: Illustration of the graph transformation process for LAMP in GraphQA task. (a) An example of two arguments with a "Support" relation from the ExplaGraphs dataset. (b) The original graph structure where relations are represented as labeled edges. (c) Our transformed graph, where each original edge is reified into a dedicated node (orange), creating a bipartite-like structure that allows LAMP to process textual relational information more effectively.

## C  BASELINE

For node classification, we compare LAMP with a range of representative methods widely adopted in prior work. In addition, for the GraphQA setting we further include G-Retriever, a recent GraphRAG designed for textual graph question answering.

- **MLP.** As a non-graph baseline, we use a simple Multi-Layer Perceptron (MLP) that operates solely on node embeddings.

- **GCN (Zhang et al., 2019).** We include Graph Convolutional Network (GCN) as a traditional GNN baseline. GCN learns node representations by iteratively aggregating features from their local neighborhoods using a spectral convolution operator.

- **GAT (Veličković et al., 2017).** A type of GNN with attention weights to differentiate neighbor importance during aggregation. This design improves robustness to noisy neighbors, making GAT a representative example of graph models that enhance aggregation through attention mechanisms.

- **GraphSAGE (Hamilton et al., 2017).** As a representative of spatial-domain GNNs, we include GraphSAGE. It learns node embeddings by directly aggregating features from sampled local neighborhoods. This spatial approach provides strong inductive capabilities, allowing the model to generalize to new nodes and graphs.

- **BERT (Devlin et al., 2019).** A widely used pre-trained bidirectional transformer encoder. Here, BERT is applied to node texts independently, ignoring graph structure. It serves as an LM-only baseline for evaluating the benefit of structural modeling.

- **RoBERTa (Liu et al., 2019).** We also include RoBERTa, an optimized successor to BERT, to serve as another LM-only baseline. RoBERTa enhances the pre-training procedure of BERT through several key modifications: it is trained on a significantly larger corpus, and removes the next-sentence prediction (NSP) objective. Similar to the BERT baseline, RoBERTa is directly applied to the node text.

- **Qwen-2.5-7B-Instruct (Team, 2024).** A modern decoder-only LLM with strong text understanding and generation capabilities. We adopt its 7B Instruct variant as the backbone in our experiments, using it both as a direct LM-based baseline and as the foundation for building LAMP.

- **OFA (Liu et al., 2024).** An LLM-GNN hybrid method that unifies diverse cross-domain data and task types by leveraging Text-Attributed Graphs (TAGs) with a single LLM encoder, and introduces a Graph Prompting Paradigm (GPP) based on Nodes-of-Interest (NOI) to enable zero-shot and few-shot classification.

- **LLaGA (Chen et al., 2024).** An LLM-centric method that combines graph with LLM by converting graph into serial text sequences using predefined templates and mapping them into the LLM embedding space with a projector.

- **PromptGFM (Zhu et al., 2025).** A recent SOTA "LLM-as-GNN" approach that summarizes neighborhoods into language prompts by GPT-4o (OpenAI, 2024), demonstrating the feasibility of LLM-based message passing under context-length constraints.

- **G-Retriever (He et al., 2024).** A GraphRAG method for textual graph question answering, which casts subgraph retrieval as a prize-collecting Steiner tree (PCST) to select a compact, query-relevant connected subgraph and generates answers with an LLM. It serves as a representative GraphQA baseline.

To align with the text-rich graph setting, BERT is used for generating node embeddings across all applicable baselines.

## D RELATED WORK

We organize the related work into three key areas that build the foundation for our proposed LAMP framework: (1) learning on text-rich graphs, which establishes the problem domain; (2) the message passing paradigm in GNNs, which inspires our core aggregation mechanism; and (3) recent efforts to integrate graphs into LLMs, which defines the current state-of-the-art and the specific gap LAMP addresses.

### D.1 TEXT-RICH GRAPHS

A wide range of real-world graphs are text-rich, including knowledge graphs, academic networks, and product graphs (Galkin et al., 2023; Zhang et al., 2023; 2024a; Yang et al., 2020). These graphs pose a unique challenge, as models must reason over both complex topological structures and long, nuanced natural language content (Kong et al., 2024). Early and still-prevalent approaches have been GNN-centric, treating text as a node attribute to be pre-processed (Jin et al., 2024). These methods, which we term **"Compressed Aggregation"**, first use a text encoder like BERT to collapse each node's text into a single, fixed-size embedding. This embedding is then propagated through a GNN (e.g., GCN (Zhang et al., 2019), GAT (Veličković et al., 2017)). While computationally efficient, this creates a severe **information bottleneck**, as the rich semantics of the original text are irreversibly lost before any graph-level reasoning occurs. Even with sophisticated joint training schemes like **GLEM** (Zhao et al., 2022) or adapter-based integration as in **GraphAdapter** (Li et al., 2023), this fundamental bottleneck persists. Another line of research directly pre-trains language models on graph-structured text to enrich node representations (Yasunaga et al., 2022; Ye et al., 2023), but these methods typically focus on enhancing the language model itself rather than defining a native message-passing mechanism for the LLM.

## D.2 MESSAGE PASSING IN GNNS

Message passing is the central paradigm of most GNNs (Feng et al., 2022; Papillon et al., 2023; Zhang et al., 2024c; Sun et al., 2024), where each node updates its representation by aggregating features from its neighbors. Foundational models like GCN (Zhang et al., 2019) and Graph-SAGE (Hamilton et al., 2017) established the effectiveness of this approach. A pivotal advancement came with the introduction of attention mechanisms in GNNs, such as GAT (Veličković et al., 2017), which weigh neighbor importance through a query–key–value formulation. This mechanism is particularly insightful for our work. For example, in GAT, a node's own representation from the previous layer, $\mathbf{h}_i^{(l)}$, serves as the **query**, while the representations of its neighbors, $\{\mathbf{h}_j^{(l)}\}_{j \in \mathcal{N}(i)}$, serve as **keys** and **values**. This asymmetric design is powerful: it allows the target node to **actively probe** its neighborhood and selectively draw in the most relevant information, rather than being passively overwhelmed by a simple sum or average of all neighbor features. This prevents the node's accumulated knowledge from being diluted by noisy or irrelevant neighbors, a critical feature for robust representation learning.

**LAMP's architecture is directly inspired by this principle.** We argue that for a text-rich node, its most potent "query" is its **full, raw text ($\mathbf{X}_i$)**, as it contains the richest semantic context to determine what information is needed from its surroundings. Correspondingly, the **compact summary tokens of its neighbors ($\mathbf{S}_j^{(l)}$)** serve as efficient "keys" and "values" that can be probed. By performing attention over its own raw text and its neighbors' summary tokens, LAMP extends this general principle to the LLM setting. It performs a focused, context-aware aggregation that extracts maximal signal from the neighborhood without the computational burden of processing the full text of every neighbor simultaneously. **This design choice is not arbitrary; it is an adaptation of a state-of-the-art GNN principle to the LLM era, positioning LAMP as a natural and powerful evolution for learning on text-rich graphs.**

## D.3 GRAPHS IN LLMS

The rise of LLMs has spurred new paradigms for graph learning (Li et al., 2024a; Jin et al., 2024; Wang et al., 2025; Guan et al., 2024). The most direct approach is **graph serialization** (Mavromatis & Karypis, 2024), where graph structures and attributes are converted into a linear text sequence for an LLM to process. This preserves text fidelity but sacrifices **structural nativity**, forcing the model to infer topological relationships from an unnatural, sequential format and struggling with scalability when node texts are long. To move beyond naive serialization, an emerging line of work attempts to make LLMs function *as* GNNs, effectively replacing the GNN's aggregation function with an LLM. **PromptGFM** (Zhu et al., 2025) uses prompts to iteratively summarize neighborhoods and generate "language-based IDs" for nodes. **LLaGA** (Chen et al., 2024) and **InstructGLM** (Ye et al., 2023) use templates to format local subgraphs into sequences. **GOFA** (Kong et al., 2024), **GraphFormers** (Yang et al., 2021), and **GL-fusion** (Yang et al., 2024) inject trainable GNN-like layers or adapters into a frozen LLM. **OFA** (Liu et al., 2024) employs a different hybrid approach, utilizing LLM to align various graph tasks into a unified format for pre-training general GNNs. **HiCom** (Zhang et al., 2024c) explicitly focuses on creating a hierarchical compression scheme to manage long text in large neighborhoods. While these methods are innovative, they share a common limitation that motivates our work: to remain computationally feasible, they all perform some form of information compression or abstraction *before or during* the message aggregation step. The LLM aggregator in these models never reasons over the explicit, raw text of its multi-hop neighbors. For example, PromptGFM (Zhu et al., 2025) reasons on summarized IDs, and HiCom (Zhang et al., 2024c) on compressed tokens. This prevents the LLM from leveraging its greatest strength—deep contextual reasoning on raw evidence—at the most critical step. This is precisely the gap our proposed LAMP seeks to fill. By employing a **dual-representation scheme**, LAMP is the first framework where message passing is natively realized within an LLM that explicitly reasons over the raw text of its immediate neighborhood, thus achieving true semantic fidelity without sacrificing the computational efficiency needed for scalable graph propagation.

# E   PROMPT TEMPLATE USED IN LAMP

We provide the prompt templates used in different tasks for reproducibility. Each template is divided into *Node Input* (the text content of a node) and *Query Input* (the task-specific instruction given to the model).

## E.1   NODE CLASSIFICATION

**Node Input.** For node classification task, we add *soft prompt* at both the beginning and the end of the node text, further guiding the model to summarize the content into a compact representation.

**Query Input.** The classification task is formulated as a multiple-choice question, where the model is asked to predict the research category of the given node. The full node text (title and abstract, wrapped with soft prompts) is also included, ensuring that the model makes predictions conditioned on the same explicit textual information as other backbones.

---

**Prompt Example for Node (Node Classification Task)**

*Above are papers related to the following paper:* [Paper_ID] Paper #446271. [Title] Mapping Bayesian Networks to Boltzmann Machines [Abstract] We study the task of tnding a maximal a posteriori (MAP) instantiation of Bayesian network variables, given a partial value assignment as an initial constraint. This problem is known to be NP-hard, so we concentrate on a stochastic approximation algorithm, simulated annealing. This stochastic algorithm can be realized as a sequential process on the set of Bayesian network variables, where only one variable is allowed to change at a time. Consequently, the method can become impractically slow as the number of variables increases. We present a method for mapping a given Bayesian network to a massively parallel Bolztmann machine neural network architecture, in the sense that instead of using the normal sequential simulated annealing algorithm, we can use a massively parallel stochastic process on the Boltzmann machine architecture. The neural network updating process provably converges to a state which solves a given MAP task. *Please summarise all the given information.*

---

**Prompt Example for Query (Node Classification Task)**

Paper #446271 has the title 'Mapping Bayesian Networks to Boltzmann Machines' and the abstract 'We study the task of tnding a maximal a posteriori (MAP) instantiation of Bayesian network variables, given a partial value assignment as an initial constraint. This problem is known to be NP-hard, so we concentrate on a stochastic approximation algorithm, simulated annealing. This stochastic algorithm can be realized as a sequential process on the set of Bayesian network variables, where only one variable is allowed to change at a time. Consequently, the method can become impractically slow as the number of variables increases. We present a method for mapping a given Bayesian network to a massively parallel Boltzmann machine neural network architecture, in the sense that instead of using the normal sequential simulated annealing algorithm, we can use a massively parallel stochastic process on the Boltzmann machine architecture. The neural network updating process provably converges to a state which solves a given MAP task.'
Question: Which category should Paper #446271 be classified as? You can select one from ['Neural_Networks', 'Case_Based', 'Theory', 'Reinforcement_Learning', 'Probabilistic_Methods', 'Rule_Learning', 'Genetic_Algorithms'].

---

## E.2   GRAPH QUESTION ANSWERING (GRAPHQA)

**Node Input.** In the GraphQA task, the node/edge texts are relatively short (as in the ExplaGraphs dataset). Therefore, we directly feed the raw node text into the model without additional soft prompts.

**Query Input.** The query is expressed in natural QA format, requiring the model to reason over the entire graph.

**Prompt Example for Node (GraphQA Task)**

created by

**Prompt Example for Query (GraphQA Task)**

Argument 1: Safe spaces are a redundant and unnecessary practice. Argument 2: Some people have no support or guidance and need it to be available for them.
Question: Do argument 1 and argument 2 support or counter each other? Answer in one word in the form of 'support' or 'counter'.

## F   GENERATION ACROSS MESSAGE-PASSING ROUNDS

To better understand what happens during the propagation process of LAMP, we compare model outputs when conditioning on the **penultimate** message-passing round KV versus the **last** round KV of message passing. We provide representative cases from the **Cora** dataset in Tab. 8, which illustrate how intermediate and final outputs can differ. Based on these observations, several consistent patterns emerge:

- Even though only the last round is trained to match the ground truth, the penultimate round already produces **non-trivial generations** that often resemble intermediate analysis.
- When the penultimate round prediction is incorrect, the final round sometimes **corrects it**, likely due to having aggregated information from a larger neighborhood scope.
- Conversely, we also observe cases where the final round prediction is **misled by** an erroneous intermediate generation.
- Overall, these qualitative results suggest that intermediate rounds are not merely placeholders for information aggregation, sometimes exhibiting reasoning-like traces before the final prediction.

## G   THE USE OF LLMS

In the preparation of this manuscript, LLMs are utilized as a general-purpose writing assistant. Its role was strictly limited to improving the grammar, clarity, and readability of the text. The LLMs are not used for research ideation, conducting experiments, or the generation of any core scientific content. The authors take full responsibility for all content presented in this paper, including any text revised with the assistance of the LLM.

Table 8: Examples showing generations from different message-passing rounds in LAMP. While only the final round is supervised during training, we observe that the penultimate round KV also produces meaningful outputs. Comparisons across rounds reveal intermediate reasoning behaviors, occasional error correction, and in some cases error propagation. *Italic text* indicates predictions that do not match the ground truth.

| Paper Title | Intermediate Output (penultimate round KV) | Final Output (last round KV) | Ground Truth |
|---|---|---|---|
| Unsupervised Learning Using MML | This paper should be classified into: *Theory methods* | This paper should be classified into: Probabilistic_Methods | This paper should be classified into: Probabilistic_Methods |
| First Order Regression | This paper should be classified into: *Theory* | This paper should be classified into: Rule_Learning | This paper should be classified into: Rule_Learning |
| First Order Regression: Applications in Real-World Domains | This paper should be in the rule-based category as it deals with rule based machine learning | This paper should be classified into: Rule_Learning | This paper should be classified into: Rule_Learning |
| Probably Approximately Optimal Derivation Strategies | This paper should be classified into: Theorys category: theory abstract: an paper describes a pao algorithm that first uses mu set of training examples to approximate these probability values and then uses these estimates to produce mu "probably approximately optimal" strategy that is given any *; ffi ¿ 0, pao produces mu strategy whose cost is within * of mu cost of mu optimal strategy with probability greater ... | This paper should be classified into: Theory | This paper should be classified into: Theory |
| A Decision Tree System for Finding Genes in DNA | This paper should be in *theory* category, it describes a decision tree system for finding genes in vertebrate dna sequences, and uses new methods for identifying start codons, donor sites, and acceptor sites, and brings them together in a frame sensitive dynamic programming algorithm that findss the minimum segmentation of a dna sequence into coding and ... | This paper should be classified into: Neural_Networks | This paper should be classified into: Neural_Networks |
| Space-Frequency Localized Basis Function Networks for Nonlinear System Estimation and Control | This paper should be classified into: Neural This paper deals with the on-line generation of nonparametric models for nonlinear systems using tools from multiresolution analysis and wavelet theory. It extendss earlier results on adaptive control and identification of nonparametric systems using wavelet basis functions to on-line generation of nonparametric models for nonparametric systems with wavelet basis functions ... | This paper should be classified into: Neural_Networks | This paper should be classified into: Neural_Networks |
| Gas Identification System using Graded Temperature Sensor and Neural Net Interpretation | This paper should be classified into: Neural_ interpretation (*or* case-based reasoning) as it uses a (fuzzy logic) for interpretation of sensor data. | This paper should be classified into: Neural_Networks | This paper should be classified into: Neural_Networks |
| Using Partitioning to Speed Up Specific-to-General Rule Induction | This paper should be classified into: *rule induction (theory)* | This paper should be classified into: *Rule_Learning* | This paper should be classified into: Case_Based |
| Genetic Algorithm based Scheduling in a Dynamic Manufacturing Environment | This paper should be classified into: Genetic Alculms | This paper should be classified into: Genetic_Algorithms | This paper should be classified into: Genetic_Algorithms |
| Combining Rules and Cases to Learn Case Adaptation | This paper should be in the case-based category. It describes a new approach to learn case adaptation knowledge in case-based reasoning (cbb) systems. The paper is not about neural networkss, theory, reinforcement learning, probabilistic methods, rule learning, or genetic algorithmss. | This paper should be classified into: Case_Based | This paper should be classified into: Case_Based |
| Associative Reinforcement Learning: Functions in k-DNF | This paper should be classified into: Reinmunl Learning | This paper should be classified into: Reinforcement_Learning | This paper should be classified into: Reinforcement_Learning |