# OpenReview forum: "LAMP: An LLM-based Message Passing Architecture for Text-Rich Graphs"
_ICLR.cc/2026/Conference — ICLR 2026 Conference Withdrawn Submission_

### Official Review · Reviewer_hkTD · 2025-10-22

**Soundness:** 2
**Presentation:** 2
**Contribution:** 2
**Rating:** 2
**Confidence:** 4

**Summary:**

In the paper, the authors focus on the rich-text graphs and propose an LLM-based architecture called LAMP to process the graphs. The architecture modify an LLM to be a node encoder and concatenate the neighbors encoding representation to learn from the graph structure. The architecture can both perform reasoning on both raw node language and graph structure. The experiments show that LAMP can finish classic graph tasks such as node classification and graphQA tasks.

**Strengths:**

1. The architecture is simple and easy to understand and implement. The LLM is less modified to be a node encoder.
2. The efficiency seems good. The architecture roughly has the same time complexity level as classic GNNs.

**Weaknesses:**

1. **Novelty overclaim**: though the architecture is new, the idea to make reasoning on the raw data is not novel. In the paper [1], the authors also (1) use an summary token provided by an LLM as the first node representation, (2) and also keep all the raw node text (and even the edge text) to perform reasoning on text-rich graph. They use an cross-attention mechanism to avoid cost of full-text propagation. Besided, their architecture can keep the permutation invariance strictly. Their reported results are also better than LAMP on 'history' dataset.
2. **Architecture design**: There are some design choices that are not well justified. For example, (1) target on the graph structure learning, the permutation invariance of neighbors order is not theoretically guaranteed. (2) At the same time, although the authors emphasize the importance of the raw text information, the final generation process cannot directly access the raw text of the nodes. The text information is also compressed into the summary tokens $S$ and cannot be directly used. (It could be incorrect because the authors do not explain the generation process in detail. For example, it is not clear what is the $K$ and $V$ in the equation 3.) (3) The message passing and summarization process is task-agnostic, the task description is only used in the final generation process. Considering different tasks may require essentially different information from the graph, it is better to have task-specific message passing and summarization process.
3. **unclear explanation of experiment results**: the experiments need further explanation. The perplexity in Table 2 cannot support the authors' claim that LAMP learn the neighbor's information becuase the perplexity is not compared to a baseline. Compared to the "self" results, does it show that LAMP do not really recover the neighbors' text? At the same time, only the 2-hop neighbor is not enough for some graphs. For example, for some knowledge graphs, the necessary information may be 3-hop or more away. The dataset used in the experiments are classic but somehow old. More recent datasets like ogbn-arxiv can be used to further validate the model's effectiveness. Also notice that the LLMs have get strong performance (like Qwen-2.5), which suggests that the structure information could be not so useful for these tasks. The Cross-Node shuffle show not significant performance drop, which also suggests that the structure information may not be well used.
4. **Insufficient experiments**: I believe the text-rich graph learning is a very important direction and more complex senarios. The citation graphs (as well as the co-purchase networks) are too naive and special. These graphs are highly homophilous and also the text have enough information to finish the tasks. There is no need to operate deeper structure learning or some complex reasoning on the graph. I believe some experiments on some heterogeneous graphs or multi graphs could better validate the model's effectiveness. For example, the multi-hop reasoning on knowledge graphs can be useful testbed.

[1] Yang et. al., GL-Fusion: Rethinking the Combination of Graph Neural Network and Large Language model. [link](https://arxiv.org/abs/2412.06849)

**Questions:**

See weaknesses. And:
1. What is the exact generation process?
2. Are there some edge-level or graph-level tasks?
3. Where is the discussion of your LLM usage?

---

> ### Author Response · Authors · 2025-11-21
> **Response to Reviewer hkTD (1/n)**
>
> _W1 Novelty overclaim, compared to GL-Fusion_
>
> We thank the reviewer for bring this up. We acknowledge GL-Fusion as a strong representative of the "Invasive Integration" paradigm (as categorized in our **General Response: Clarification on Novelty and Contributions**). However, we would like to clarify that while both methods leverage summary tokens and raw text, they differ fundamentally in architectural philosophy and the usage of raw text:
>
> 1. **Difference in Architecture: Invasive Integration vs. Behavioral Simulation**
>
> **GL-Fusion (Invasive Integration):** It typically maintains separate modules—Structure-Aware Transformers (to capture structure) and linked raw text of node via a cross-attention mechanism. The structural reasoning is partially offloaded to the additional module.
>
> **LAMP (Behavioral Simulation):** LAMP eliminates the need for auxiliary structure-learning module. We architecturally recast the LLM decoder itself to function as the graph kernel. The message passing and aggregation is natively executed via the decoder's self-attention, offering a more unified and simplified blueprint.
>
> 2. **Difference in The Role of Raw Text: Passive Refinement vs. Active Guidance**
>
> In contrast to GL-Fusion, where raw text serves as a **passive context** for node's final state refinement, LAMP assigns raw text a distinct and active role **in the message-passing step**. In LAMP, the raw text serves as the high-fidelity Query during the aggregation of message (Eq. 2). This structural difference is critical: GL-Fusion performs neighbor weighting and selection in a compressed vector space before considering the full semantics, thereby failing to exerting the LLM’s native linguistic mastery to guide the aggregation process itself. LAMP, **conversely**, achieves raw-text-guided aggregation, ensuring that the selection and interpretation of neighbor messages are dynamically driven by the target node’s full semantic content, effectively resolving the Anchor Node Bottleneck within a unified LLM architecture.
>
> 3. **Additional: Invariance and Performance**
>
> > Besided, their architecture can keep the permutation invariance strictly. Their reported results are also better than LAMP on 'history' dataset.
> >
>
> **Invariance:** We agree that the design of GL-Fusion (Permutation Invariant Causal Self-Attention) has strict invariance. However, as shown in our Table 6, LAMP also successfully learns this invariance from data (showing negligible variance under permutation), achieving the practical requirement for graph learning without constraining the architecture to non-transformer components.
>
> **Performance:** We acknowledge the impressive results reported in GL-Fusion's original paper [R5] on the 'History' dataset. However, we note that a direct numerical comparison may not be strictly apples-to-apples due to differences in data splits.
>
> ****
>
> [R5] Gl-fusion: Rethinking the combination of graph neural network and large language model. 2024

---

> ### Author Response · Authors · 2025-11-21
> **Response to Reviewer hkTD (2/n)**
>
> _W2&Q1 Architecture design_
>
> Thank you for your detailed questions. We appreciate this opportunity to clarify our design choices below.
>
>
>
> > _(1) target on the graph structure learning, the permutation invariance of neighbors order is not theoretically guaranteed._
> >
>
> We acknowledge that unlike standard GNNs (which use pooling operators during aggregation), the attention mechanism in LLMs is not theoretically permutation-invariant due to positional encodings. Therefore, to verify whether LAMP can learn to approximate invariant functions from data, we conducted the Neighbor-Order Shuffle experiment (Table 6). As detailed in our response to Reviewer 3RsJ's Q1, randomly permuting the input order of neighbors resulted in negligible performance variance (e.g., 84.87% vs. 84.81% on Cora). This provides strong empirical evidence that LAMP learns to treat the neighborhood as a set, satisfying the practical requirement for graph learning.
>
>
>
> > _(2) At the same time, although the authors emphasize the importance of the raw text information, the final generation process cannot directly access the raw text of the nodes. The text information is also compressed into the summary tokens and cannot be directly used. (It could be incorrect because the authors do not explain the generation process in detail. For example, it is not clear what is the and in the equation 3.)_
> >
>
> We would like to clarify three key points:
>
> + **Clarification on Generation (Equation 3):** Following the FocusLLM [R6] paradigm, the generation is conditioned on the Key-Value (KV) pairs derived from the final-layer summary tokens. Specifically, in Eq. 3, $Q$ represents the task query embeddings, while $K$ and $V$ are the concatenated KV caches of the summary tokens from relevant nodes.
> + **Information Bottleneck (KV Cache vs. Embedding):** While the final generation indeed uses summary tokens, we argue that a sequence of summary tokens (and their corresponding high-dimensional KV caches) has a significantly higher information bandwidth than the single fixed-size vector used in traditional GNNs, as described in [R6] and [R7].
> + **Importance of Raw Text:** Our claim of "reasoning over raw text" focus on the Message Passing Phase, not the final readout. During the $L$ layers of propagation, the update of summary tokens is directly attended to the target node's raw text (see Eq. 2). This ensures that the final summary tokens—which serve as the input for generation—are produced via a high-fidelity, text-guided aggregation process, retaining far more semantic detail than summaries produced by vector-only aggregation.
>
>
>
> > _(3) The message passing and summarization process is task-agnostic, the task description is only used in the final generation process. Considering different tasks may require essentially different information from the graph, it is better to have task-specific message passing and summarization process._
> >
>
> We agree that message passing should be task-aware and we are indeed aware of it during the model design. While the initialization (pre-training) is task-agnostic (node text reconstruction), the fine-tuning process adapts the message passing behavior to the downstream task by employing LoRA (Low-Rank Adaptation) on the weights of the decoder during fine-tuning (see 2.4 TRAINING RECIPE in the manuscript). Since the message passing is natively executed by the decoder's attention mechanism, optimizing the LoRA parameters against the task-specific objective automatically teaches the model how to summarize and aggregate neighbors specifically for that task.
>
> ****
>
> [R6] FocusLLM: Precise Understanding of Long Context by Dynamic Condensing. 2024
>
> [R7] Long Context Compression with Activation Beacon. ICLR 2025

---

> ### Author Response · Authors · 2025-11-21
> **Response to Reviewer hkTD (3/n)**
>
> _W3 further explanation of experiment results_
>
> > _(1) The perplexity in Table 2 cannot support the authors' claim that LAMP learn the neighbor's information becuase the perplexity is not compared to a baseline. Compared to the "self" results, does it show that LAMP do not really recover the neighbors' text? At the same time, only the 2-hop neighbor is not enough for some graphs. For example, for some knowledge graphs, the necessary information may be 3-hop or more away._
> >
>
> We would like to clarify the methodology and interpretation of our perplexity experiments and address the points raised.
>
> 1. **Lack of a Baseline:**
>
> The crucial baseline was indeed provided. As stated in the caption of Table 2 (Line 309-312), "Qwen2.5-7B-Instruct (21.72*) is the backbone reference; its value is the raw language modeling perplexity obtained by directly continuing node text."
>
> Specifically, the untrained backbone LLM's perplexity (21.72) represents the "zero-shot" or naive ability to reconstruct/predict text. After pre-training, our LAMP model achieves an apparent lower perplexity across various settings presented in Table 2. The drop in perplexity is direct evidence that the model learnt meaningful information through the compression process, enabling it to reconstruct the text far better than by chance or raw language modeling alone.
>
> 2. **"Self" vs. "Neighbor":**
>
> The "Self" task is much easier, as it reconstructs text that is fully present in the input before compression. The "Neighbor" task is inherently harder, as it requires the model to imply the node content from its neighbors' compressed information. Therefore, "Neighbor" mode's higher perplexity (than "Self" mode) is expected and does not indicate failure since the perplexity for LAMP<sub>mp=1</sub> is still lower than that of the baseline (21.72).
>
> 3. **On the Sufficiency of 2-Hop Neighbors:**
>
> We agree that for some task, information from 3-hop or more distant neighbors can be more useful. Our study focused on 1- and 2-hops to validate the feasibility and effectiveness of our core method. Crucially, our state-of-the-art performance on the downstream QA task (Table 4) demonstrates that the information learned from just 2-hop neighbors is already effective and sufficient to outperform the baselines. This provides compelling evidence for the utility of our approach, even with a limited hop-size. We have acknowledged this in the Limitations section (Line 475-481) and leave the exploration of deeper architectures (which require extensive pre-training) to future work.
>
>
>
> > _(2) The dataset used in the experiments are classic but somehow old. More recent datasets like ogbn-arxiv can be used to further validate the model's effectiveness._
> >
>
> As suggested, we have extended our evaluation to include ogbn-arxiv, a widely recognized modern benchmark. We have provided the detailed experimental setting and full results in our Response to Reviewer 2kRn (W1). As shown in Table 3 of the revision (and Table T1 in our response for Reviewer 2kRn), LAMP achieves highly competitive performance (e.g., 75.38% accuracy) on this dataset, confirming LAMP’s architectural advantages.
>
>
>
> > _(3) Also notice that the LLMs have get strong performance (like Qwen-2.5), which suggests that the structure information could be not so useful for these tasks._
> >
>
> We agree that modern LLMs are strong semantic learners. However, we argue that LAMP’s performance is significant precisely because it surpasses a maximized text-only baseline.
>
> + **Rigorous Baseline:** We fine-tuned the Qwen-2.5 backbone with a LoRA rank of 64, much higher than the standard rank (typically 8 or 16). This was intentionally designed to push the text-only performance to its theoretical ceiling.
> + **Structural Utility:** Even against this competitive baseline, LAMP consistently outperforms it across all datasets. This proves that LAMP successfully extracts complementary structural signals that even a well-optimized LLM cannot infer from isolated text alone.
>
>
>
> > _(4) The Cross-Node shuffle show not significant performance drop, which also suggests that the structure information may not be well used._
> >
>
> We have addressed this concern in detail in our Response to Reviewer 3RsJ (Q1). To sum up, through comparison, LAMP remains stable under Neighbor-Order Shuffle (invariance) but consistently degrades when topology is broken via Cross-Node Shuffle (sensitivity), proving its structural awareness. The small absolute drop simply reflects the text-dominant nature of the dataset, where raw text provides the primary signal.

---

> ### Author Response · Authors · 2025-11-21
> **Response to Reviewer hkTD (4/n)**
>
> _W4 The citation graphs (as well as the co-purchase networks) graphs are highly homophilous and also the text have enough information to finish the tasks. Need experiments on some heterogeneous graphs or multi graphs_
>
> We thank the reviewer for this excellent point. While our current experiments focus on homophilous graphs, LAMP is designed to be readily extensible to heterogeneous graphs.
>
> The extension is **straightforward**: we would employ a type-aware prompt to generate the summary token for each node type. To be specific, prompt would be structured to explicitly inform the LLM of the node or edge types of the neighbors. For instance, the prompt would differentiate between neighbors that are "authors," "cited papers," or "publication venues," feeding their corresponding text into categorized fields.
>
> This approach requires no architectural changes to LAMP, highlighting the flexibility of its non-invasive, prompt-centric design. We have added the empirical evaluation on heterogeneous graphs as our immediate future work and have updated the Conclusion section (Lines 478-480) to reflect it.
>
> _Q2 Are there some edge-level or graph-level tasks?_
>
> We would like to clarify that our evaluation already includes a graph-level task, and explain why edge-level tasks were not the primary focus in the original manuscript.
>
> 1. **Graph-Level Task (GraphQA):** We would like to point out that the GraphQA experiment (Sec 3.2.1) is structurally equivalent to Graph Classification. In the experiment, LAMP aggregates information across the entire topology (via KV cache pooling of all nodes) to determine a global label. LAMP achieves 93.86% (Table 4), outperforming specialized baselines, which confirms its strong global reasoning capabilities.
> 2. **Edge-Level Tasks (e.g., Link Prediction):** We argue that unlike "Graph Foundation Models" aiming for universal coverage, LAMP targets High-Fidelity Reasoning on text-rich structures. Since Node Classification (Local) and GraphQA (Global) already bracket the spectrum of reasoning granularities, they sufficiently validate the architecture's efficacy. We thus prioritized these primary use cases. However, we fully recognize the importance of edge-level tasks and plan to include explicit link prediction benchmarks in future work to further broaden LAMP's application scope.
>
> _Q3 Where is the discussion of your LLM usage?_
>
> We did include a discussion on the use of LLMs in the original manuscript in Appendix G.

---

> ### Author Response · Authors · 2025-11-27
>
> Dear Reviewer hkTD,
>
> Thank you for your thoughtful feedbacks. As the discussion period is coming to end soon, we would like to ask if there are any remaining points where you feel further clarification would be beneficial. We'd be grateful for a chance to hear your thoughts.
>
> Best Regards,
>
> Authors of Paper 2983

---

### Official Review · Reviewer_Q53Q · 2025-10-25

**Soundness:** 2
**Presentation:** 2
**Contribution:** 2
**Rating:** 4
**Confidence:** 4

**Summary:**

The authors propose several key desiderata for their ideal text-rich graph model and introduce the LAMP model, featuring an innovative architecture: the dual-representation scheme. In each message-passing layer, LAMP’s aggregator (based on an LLM) simultaneously attends to two types of information: the full raw text of the target node and compact summary tokens from its neighboring nodes. The LLM acts as a rewriter in a different form, generating the messages to be passed—all in textual format.

**Strengths:**

1. It balances semantic fidelity and structural integrity.

2. It unifies node classification (a discriminative task) and graph question answering (a generative task) under a single generative paradigm, enabling task-agnostic reasoning through a KV cache mechanism.

**Weaknesses:**

The approach is overly naive. The authors’ method can be summarized as follows: the LLM generates a summary based on neighbor information, and this summary is then passed as the message to produce summaries for the next layer. This kind of approach has already become ubiquitous in 2024.

The experimental comparisons are insufficient; the paper lacks comparisons against more advanced models such as TAPE, LangTopo, LLaGA, and UniGraph.

**Questions:**

I am genuinely curious: where do the authors believe the novelty or interesting aspect of their design originates? This seems more like a summary model than a genuinely new text-rich graph model. ::)

---

> ### Author Response · Authors · 2025-11-21
> **Response to Reviewer Q53Q**
>
> _W1&Q1 The approach is overly naive / the novelty or interesting aspect of their design originates_
>
> While we provide a comprehensive architectural positioning of LAMP compared to existing paradigms (via the Information Bottleneck framework) in our **General Response: Clarification on Novelty and Contributions**, we specifically address your concerns regarding the "naive" nature and ubiquity of the design here:
>
> 1. **Summarization IS the Essence of Graph Learning** We argue that iterative neighborhood summarization is, by definition, the fundamental mechanism of modern Graph Neural Networks (MPNNs) [R4]. Classical GNNs (e.g., GAT) generate a "summary" using MEAN or SUM operators. LAMP adheres to this basic paradigm but upgrades the kernel to an LLM Decoder for semantic-aware summarization. The "simplicity" of mapping GNN operations to LLM primitives is an intentional design choice to create a unified architecture.
> 2. **Mechanism: Raw-Text-Guided Aggregation** Unlike ubiquitous summary models which are often input-agnostic, LAMP implements an asymmetric attention mechanism (Eq. 2). The Target Node's raw text ($X_i$) serves as a high-fidelity Query to attend to Neighbors' Summaries ($S_j$). This aligns well with the context-aware aggregation logic of GNNs (where node state queries neighbors) within the LLM space.
>
>
>
> _W2 The experimental comparisons are insufficient_
>
> We have addressed the comparative evaluation in detail in our Response to Reviewer 2kRn (W1 & W2). In the revision, we also expanded our experiments to the new ogbn-arxiv dataset and incorporated LLaGA (as you suggested) and OFA (as a representative graph foundation model) as key baselines. As shown in Table 3 (and Table T1 in our response to Reviewer 2kRn), LAMP achieves highly competitive performance against these recent approaches (e.g., 75.38% on ogbn-arxiv).
>
> ****
>
> [R4] Neural Message Passing for Quantum Chemistry. 2017

---

> > ### Comment · Reviewer_Q53Q · 2025-11-28
> >
> > Thanks

---

> ### Author Response · Authors · 2025-11-27
>
> Dear Reviewer Q53Q,
>
> Thank you for your thoughtful feedbacks. As the discussion period is coming to end soon, we would like to ask if there are any remaining points where you feel further clarification would be beneficial. We'd be grateful for a chance to hear your thoughts.
>
> Best Regards,
>
> Authors of Paper 2983

---

### Official Review · Reviewer_3RsJ · 2025-11-01

**Soundness:** 2
**Presentation:** 3
**Contribution:** 1
**Rating:** 2
**Confidence:** 3

**Summary:**

This work proposes, LAMP, a new architecture combining GNN and LLM for graph with text attributes. This work aggregate information from neighbors for each node in decoder layer level. Therefore, it do not need to compress node text with a seperate LLM. Experiments show that LAMP works well on node classification tasks.

**Strengths:**

1. Clear illustration of LAMP architecture.

**Weaknesses:**

1. Insufficient related work: [1] is missing.
2.  The architecture is very similar to GOFA:: they both iteratively compress GNN text information into memory tokens and passing memory tokens between nodes only.
3. Insufficient experiment. The baseline combining GNN and LLM only includes PromptGFM in the main table, and representative like LlaGA, GLEM, GOFA cited in this work are not included. Experiments only contains small node classification tasks and one GraphQA dataset. While larger datasets like ogbn-arxiv and ogbn-products, and link prediction tasks (like tasks used in GLEM and GOFA) are not included.  All these making the claims of scalabillity to number of nodes and strong performance not persuasive.
4. The scalability experiments do not include other models as baseline.

[1] One For All: Towards Training One Graph Model For All Classification Tasks. ICLR 2024

**Questions:**

1. In Table 6, Cross Node shuffle leads to nearly no performance decrease. Does this result mean the model do not use graph structure informance at all?

---

> ### Author Response · Authors · 2025-11-21
> **Response to Reviewer 3RsJ**
>
> _W1 Insufficient related work: OFA_
>
> Thank you for the thoughtful suggestion. We agree that OFA is a relevant baseline and have updated our manuscript accordingly.
>
> + **Related Work:** A discussion of OFA's methodology is now included in our Related Work (Appendix D.3).
> + **Experimental Results:** We have conducted new experiments to benchmark against OFA. The updated results in Table 3 show that our method, LAMP, consistently outperforms OFA in most datasets, under the identical experimental setting. For example, LAMP achieves an average improvement of 3.89% in Acc over OFA.
>
> We believe this new comparison provides a more comprehensive evaluation and further validates the effectiveness of LAMP.
>
> _W2 The architecture is very similar to GOFA_
>
> While both LAMP and GOFA employ iterative processing (the standard paradigm in Graph Neural Networks) and memory tokens (a prevailing technique in Long-Context Compression) to manage context, we would like to clarify that they represent fundamentally distinct paradigms for information aggregation.
>
> As detailed in our **General Response: Clarification on Novelty and Contributions**, the distinction lies in how the aggregation is performed and what information is accessible during message passing:
>
> + **Invasive Integration (GOFA) vs. Behavioral Simulation (LAMP):**
>
> **GOFA** follows an "Invasive Integration" paradigm: It interleaves trainable GNN layers into the LLM backbone. In other words, the structural aggregation is largely handled by these auxiliary modules, which typically operate on compressed embeddings. **LAMP**, in contrast, adopts a "Behavioral Simulation" manner: LAMP do not rely on auxiliary GNN-like modules. Instead, we architecturally recast the LLM decoder itself to function as the graph kernel.
>
> + **Resolution of the Anchor Node Bottleneck:**
>
> **In GOFA**, the aggregation module typically operates on nodes vectors. Even if memory tokens are passed, the aggregation does not explicitly reason over the target node's raw text during message passing. This is the "Anchor Node Bottleneck" we identify in the revised Introduction. However, **In LAMP**, we implement raw-text-guided Aggregation. At each message-passing step, the update is uniquely conditioned on the uncompressed raw text of the target node (as a high-fidelity query), achieving a better "text-structure unification" that GOFA's modular design cannot support.
>
> In summary, while both use tokens for propagation, GOFA uses them to feed an external structural module, whereas LAMP uses them to enable the LLM to perform native, raw-text-guided message passing.
>
>
>
> _W3 Insufficient experiment_
>
> We agree that evaluating on larger-scale benchmarks and recent baselines is crucial. We have addressed this in detail in our Response to Reviewer 2kRn (W1 & W2). In the revision, we added results on ogbn-arxiv and compared with OFA and LLaGA. As shown in Table 3 (and Table T1 in Response for Reviewer 2kRn's), LAMP is highly competitive with these baselines (e.g., 75.38% on ogbn-arxiv).
>
> _W4 The scalability experiments do not include other models as baseline_
>
> The goal of the scalability experiment was to verify LAMP's linear complexity, addressing the critical "context explosion" bottleneck inherent to native LLM reasoning paradigms (e.g., PromptGFM). So, we focused the comparison on the Naive LLM-as-GNN approach to demonstrate this architectural breakthrough.
>
> + **Comparison with Naive LLM-as-GNN** PromptGFM performs reasoning over raw text typically concatenate all neighbor texts into the context window. This results in Quadratic Complexity $O(L_{total}^2)$ regarding input length.
> + **Comparison with other hybrids (e.g., GOFA, LLaGA, OFA)**: We acknowledge that invasive/adapter-based architectures are highly efficient. We would like to point out that the objective was not to race against lightweight adapters in absolute speed, but to empirically confirm that LAMP also achieves linear scalability ($O(N)$). This proves that LAMP successfully makes native, raw-text-guided reasoning computationally feasible for large neighborhoods, narrowing the gap between "naive/unscalable" and "invasive/efficient" paradigms.
>
> _Q1 Explanation for Cross Node shuffle performance's Table 6_
>
> We would like to clarify that the experimental results actually verify LAMP’s structural learning capability through the contrast between the two settings (**Neighbor-Order Shuffle** and **Cross-Node Shuffle**), rather than the absolute magnitude of the performance degradation itself:
>
> + **Invariance vs. Sensitivity:** As shown in Table 6, LAMP is more robust to Neighbor-Order Shuffle but consistently degrades under Cross-Node Shuffle, even taking the standard deviation into account. If the model ignored structure, both settings would act as indistinguishable random noise.
> + **Text-Dominance:** The small absolute drop (~0.55%) reflects the text-rich nature of the selected datasets, where raw text provides the dominant signal.

---

> ### Author Response · Authors · 2025-11-27
>
> Dear Reviewer 3RsJ,
>
> Thank you for your thoughtful feedbacks. As the discussion period is coming to end soon, we would like to ask if there are any remaining points where you feel further clarification would be beneficial. We'd be grateful for a chance to hear your thoughts.
>
> Best Regards,
>
> Authors of Paper 2983

---

### Official Review · Reviewer_2kRn · 2025-11-02

**Soundness:** 3
**Presentation:** 3
**Contribution:** 2
**Rating:** 4
**Confidence:** 4

**Summary:**

The paper proposes an LLM-based message passing scheme for textual graphs where each node always keeps its full text while only compressed neighbor tokens are passed. This aims to balance text fidelity, multi-hop propagation, and cost. Experiments show consistent gains over GNN and text-only baselines, and the structural tests indicate the model actually uses the graph.

**Strengths:**

1. The explored task of representation learning on text rich graphs without losing either semantics or structure is important; using an LLM decoder itself as the message passing engine is a natural and well motivated direction.
2. The proposed LAMP architecture is reasonable with raw text retention, summary token exchange, and iterative updates each supporting the next one.
3. The reported improvements seem to be strong, together with the structure perturbation tests, indicate that the method is truly exploiting graph signals.

**Weaknesses:**

1. The empirical scope is still narrow for a method that targets large text-rich graphs. All five classification datasets are relatively old, small datasets, and the largest is below fifty thousand nodes. It would be good to include results on larger textual graphs like OGB datasets, to stress test the paper's claims. Right now, the experiments show the method works, but only on fairly narrow settings.
2. Comparisons to the most recent LLM as GNN or long context compression approaches are not all there. PromptGFM is quoted but is not run in the same setup and uses GPT 4o; so it is more of a reference than a baseline. A few recent hybrid systems also propagate compressed activations or do hierarchical condensation [1,2]; those should be run in the same subgraph setting to make the advantage of LAMP fully convincing.
3. There is no ablation study or parameter analysis in this paper. It would be better to analyze the ratio, number of summary tokens, or placement of self tokens in the input sequence, which are central to the model design.

[1] Hierarchical Compression of Text-Rich Graphs via Large Language Models. 2024
[2] OpenGraph: Towards Open Graph Foundation Models. EMNLP 2024

**Questions:**

Please see the weaknesses

---

> ### Author Response · Authors · 2025-11-21
> **Response to Reviewer 2kRn**
>
> _W1 Include results on larger textual graphs like OGB datasets_
>
> We agree that evaluating on larger-scale benchmarks is important. In the revision, we have included results on ogbn-arxiv [R1].
>
> **Experimental Setting:** Given the strict time constraints of the rebuttal and the high computational cost of LLM fine-tuning, we sampled a representative subset of ogbn-arxiv for this comparison. Specifically, we randomly sampled _10,000_, _2,000_, and _10,000_ of the nodes from the original dataset split for training, validation, and testing, respectively. While we downsampled the target nodes for supervision, applicable models are access to the original graph topology. To ensure a fair comparison, all baselines (including the newly added baselines: OFA [R2] and LLaGA [R3]) were re-trained and evaluated on this identical subset.
>
> **Results:** As shown in the following table (and also see Table 3 in the revised paper), even on this subset, the performance trend remains consistent with our main experiments. As observed, LAMP achieved 75.38% accuracy, outperforming both LLaGA (70.55%) and OFA (64.93%).
>
> Table T1.
>
> |  | **Cora** | **Citeseer** | **Pubmed** | **History** | **Photo** | **Arxiv** |
> | :---: | :---: | :---: | :---: | :---: | :---: | :---: |
> | OFA | 81.73 | 74.19 | 86.91 | 81.75 | **77.23** | 64.93 |
> | LLaGA | 82.28 | 73.54 | 83.89 | 82.54 | 75.15 | 70.55 |
> | **LAMP** | **84.87** | **74.83** | **93.68** | **85.09** | 76.21 | **75.38** |
>
>
> _W2 Most recent approaches_
>
> To provide a more comprehensive evaluation, we have compared LAMP with two recent representative baselines during the rebuttal phase: OFA and LLaGA, under the identical experimental setting (see Table T1 in response for _W1_ or Table 3 in the revised paper). As shown in the results, LAMP surpasses OFA and LLaGA in most datasets, verifying its superiority.
>
> **Note on HiCom and OpenGraph:** We carefully considered the reviewer's suggestion to compare with HiCom and OpenGraph.
>
> + **HiCom:** Unfortunately, the official code and dataset for HiCom are not publicly available at the time of this rebuttal, making it not feasible for a faithful re-implementation of its intricate design.
> + **OpenGraph:** To represent the LLM-GNN Integration paradigm, we chose to add OFA instead of OpenGraph because its existing evaluation in a supervised fine-tuning setting (while OpenGraph's main experiments are conducted under zero/few-shot learning) provides a more direct and fair comparison to our work.
>
> _W3 additional ablation study or parameter analysis (the ratio, number of summary tokens, or placement of self tokens in the input sequence)_
>
> We agree that further ablation study will strengthen the paper. We would like to first clarify that we have performed basic ablation studies on the Compression Ratio and Message Passing Rounds. Additionally, we provide a detailed justification for the choice of token placement here.
>
> 1. **Ablation on Ratio & Token Number (Existing Results)** Table 2 (Section 3.1) already presents an ablation study on the **Compression Ratio** ($\rho$)—which directly determines the number of summary tokens ($n=\lceil\rho \cdot L\rceil$)—and the number of **Message Passing Rounds** ($mp$). According to the results, increasing the ratio from $\rho=0.05$ to $\rho=0.1$ significantly reduces perplexity under self-reconstruction mode ($11.36 \to 9.17$). This empirical evidence guided our design choice to use $\rho=0.1$ for downstream tasks, which balances semantic fidelity with computational efficiency.
> 2. **Placement of Self Tokens** In fact, we purposely placed the target node's self-tokens ($S_i^{(l)}$) at the end of the sequence, after the neighbor summaries (see Eq. 2). This design utilizes the inherent property of autoregressive LLMs: by placing the self-token last, it forces the attention mechanism to treat the preceding neighbor tokens and its raw text tokens as "context" to be integrated into the current state, ensuring the update is conceptually a "refinement" of the self-representation.
>
> Meanwhile, we are currently running these additional parameter sensitivity analyses for pre-training (e.g., testing $\rho=0.01, 0.03$ or varying token positions) to address your comments. We will update the paper again once the experiments are completed.
>
> ****
>
>
>
> [R1] Open graph benchmark: Datasets for machine learning on graphs. NeurIPS 2020
>
> [R2] One for all: Towards training one graph model for all classification tasks. ICLR 2024
>
> [R3] Llaga: large language and graph assistant. ICML 2024

---

> ### Author Response · Authors · 2025-11-27
>
> Dear Reviewer 2kRn,
>
> Thank you for your thoughtful feedbacks. As the discussion period is coming to end soon, we would like to ask if there are any remaining points where you feel further clarification would be beneficial. We'd be grateful for a chance to hear your thoughts.
>
> Best Regards,
>
> Authors of Paper 2983

---

### Author Response · Authors · 2025-11-21
**Beginning of Reply**

Thank you for your valuable feedback! We believe that addressing the reviewer’s comments has enhanced the clarity and presentation of the paper's contributions, raising it to a higher standard. The reviewer's remarks are in _italics_, followed by our responses. Moreover, in the revised version of our paper, we mark all newly added or changed paragraphs in blue. Unless otherwise noted, all references to pages, equations, sections, and citations pertain to the revised version.

---

### Author Response · Authors · 2025-11-21
**General Response: Clarification on Novelty and Contributions**

We thank the reviewers for their constructive feedback regarding the novelty of LAMP. We realize that our initial manuscript did not fully illustrate the fundamental gap LAMP fills. To address this, we have completely rewritten the Introduction to frame our contribution and we summarize the key points below:

**1. The Trilemma and the Landscape** We argue that existing paradigms for text-rich graphs face a "Trilemma" between 1) Semantic Fidelity, 2) Structural Integrity, and 3) Computational Scalability. Before an in-depth analysis of their limitations, we first distinguish between two popular LLM-GNN integration architectural paradigm:

+ **Invasive Integration:** Apporaches such as GOFA and GL-Fusion mechanically inject auxiliary structure-learning modules (e.g., GNN-like layers) into the LLM backbone. The resulting disjointed system makes the structural aggregation decoupled from the LLM's native reasoning.
+ **Behavioral Simulation:** Methods including PromptGFM and HiCom attempt to make the LLM itself execute message passing within its textual space. LAMP belongs to this more promising paradigm.

**2. The Critical Gap: Drawbacks of Current "Behavioral Simulation"** We identify that prior "Behavioral Simulation" attempts fail to resolve the above Trilemma because they impose information bottlenecks for Semantic Fidelity at the following stages:

+ **The Message Bottleneck (happens in Neighbor Transmission):** Methods relying on hard prompts (e.g., PromptGFM) compress rich neighbor context into discrete, coarse-grained natural language summaries, resulting in significant information loss.
+ **The Anchor Node Bottleneck (happens in Message Aggregation):** More seriously, all prior methods (including both Invasive Integration models like GL-Fusion and Behavioral Simulation models like HiCom) force the LLM to operate on a compressed representation of the target node itself during the aggregation step. This limits the LLM’s most powerful asset—the target node's raw text—at the most critical moment of reasoning.

**3. LAMP’s Unique Contribution** LAMP is the first architecture to dismantle these bottlenecks via raw-text-guided message passing. Unlike symmetric compression methods, LAMP employs an asymmetric dual-representation scheme:

+ **Soft Prompts from Neighbors:** We use differentiable soft prompts to propagate neighbor information, mitigating the discrete Message Bottleneck.
+ **Raw-Text from Anchor Node:** We architecturally recast the LLM decoder to use the target node’s full, uncompressed raw text as a dynamic query during aggregation.

Thus, LAMP enables native, semantically faithful message passing, which fundamentally distinguishs it from prior works that rely on pre-compressed embeddings or summaries. Specifically, the instantiation of GNN's message passing paradigm (Eq. 1) in LAMP is reframed by Eq. 2.

$
\begin{equation}
\mathbf{h}\_i^{(l+1)} = \textbf{Update}
\Big(
\mathbf{h}\_{i}^{(l)},
{\textbf{Aggregate}
(\\{
\textbf{Msg}(\mathbf{h}\_{j}^{(l)}):
j \in \mathcal{N}(i)
\\})
}
\Big),
\tag{1}
\end{equation}
$

$ \begin{equation}
    \mathbf{S}\_i^{(l+1)} = \textbf{Decoder}\Big(\big[
    \underbrace{\mathbf{S}\_{j\_1}^{(l)} \|| \cdots \|\| \mathbf{S}\_{j\_m}^{(l)}}\_{\text{neighbors}}
    \|\|
    \underbrace{\mathbf{X}\_i}\_{\text{raw text}}
    \|\|
    \underbrace{\mathbf{S}\_i^{(l)}}\_{\text{self}}
    \big]
    \Big) , j_1, ..., j_m \in \mathcal{N}(i).
    \tag{2}
\end{equation} $

---

### Note · Authors · 2026-01-06

I have read and agree with the venue's withdrawal policy on behalf of myself and my co-authors.